# Mono-Sized Anion-Exchange Magnetic Microspheres for Protein Adsorption

**DOI:** 10.3390/ijms23094963

**Published:** 2022-04-29

**Authors:** Zhe Wang, Wei Wang, Zihui Meng, Min Xue

**Affiliations:** 1School of Chemistry and Chemical Engineering, Beijing Institute of Technology, Beijing 102488, China; wangzhe@bit.edu.cn (Z.W.); wangwei2020_ripp@126.com (W.W.); mengzh@bit.edu.cn (Z.M.); 2Academy of National Food and Strategic Reserves Administration, Beijing 100037, China

**Keywords:** magnetic microspheres, surface embedding, magnetic separation, protein adsorption

## Abstract

In this study, mono-sized anion-exchange microspheres with polyglycidylmethacrylate were engineered and processed to introduce magnetic granules by penetration–deposition approaches. The obtained magnetic microspheres showed a uniform particle diameter of 1.235 μm in average and a good spherical shape with a saturation magnetic intensity of 12.48 emu/g by VSM and 12% magnetite content by TGA. The magnetic microspheres showed no cytotoxicity when the concentration was below 10 μg/mg. The magnetic microspheres possess respective adsorption capacity for three proteins including Bovine albumin, Hemoglobin from bovine blood, and Cytochrome C. These magnetic microspheres are also potential biomaterials as targeting medicine carriers or protein separation carriers at low concentration.

## 1. Introduction

The purified proteins play a crucial role in the research of protein on life activities, such as catalytic metabolic reactions and growth control. However, current separation and purification methods are tedious and time-consuming [1,2,3], such as affinity chromatography, dialysis, salting, and ultrafiltration. Magnetic separation technology has potential in protein purification due to its advantages of easy operation and rapid separation [4,5,6]. Magnetic microspheres are composite material particles [6,7] consisting of both inorganic magnetic materials providing magnetism and organic active functional groups carrying affinity ligand to target on the surface.

Magnetic microspheres have been successfully used for the separation of proteins [8,9,10] based on the interaction between protein and functional groups or special ligands on the microspheres, including electrostatic adsorption and specific adsorption. Moreover, magnetic microspheres modified with affinity ligands may have high selectivity to the target proteins, but the available ligands are limited and relative expensive [11,12]. Some commercial magnetic beads modified with monoclonal antibodies were successfully used for target substances identification, especially in diagnosis. However, the high cost, the tedious modification process, and the difference in separation effect severely limit their widespread application.

Magnetic polymer microspheres could be synthesized by several methods. The embedding method [13] is typically applied in the preparation of magnetic microspheres with a magnetic shell, which is simple and easy to carry out, but results in the magnetic particles with irregular shapes and polydisperse states. The emulsion polymerization method [14] provides monodispersed magnetic microspheres, but the small grain size beads below l.0 μm that exhibit higher separation efficiency under magnetic field are hard obtain. An in situ method [15] is a way to obtain the magnetic nanocomposite materials by binding nanoscale magnetic materials on the pre-synthetic polymer surface. During the magnetization process, the particle size and distribution of the monodisperse polymer microspheres could be maintained. Each microsphere, containing the same concentration of magnetic particles, ensures that it has uniform magnetic response in the magnetic field.

In this study, two kinds of anion-exchange microspheres were prepared by an in situ method and applied for protein adsorption study. A novel method for modification of amino-microsphere to carboxyl-microsphere by EDC, NHS, and sodium carboxymethyl cellulose was proposed. The particle size, functional groups, and magnetic properties of the resultant magnetic particles were characterized. The maximum binding capacity was relatively high compared with similar research [16,17,18].

## 2. Results and Discussion

### 2.1. Synthesis of Anion-Exchange Magnetic Microspheres

The functional magnetic microspheres were synthesized by an in situ synthesis method with moderate size and functional group beneficiation on the surface for further modification. Glycidyl methacrylate (GMA) was selected as the basic monomer to structure monodisperse polymeric microspheres. The PGMA microsphere surface was rich in amino group after reacting with EDA (Figure 1a). The amino group is a strong polar group, which can form an ionic bond and a complex coordinate bond with a metal ion, thereby reducing the probability of collision between the particles and preventing excessive aggregation of the particles. Magnetic microspheres were prepared when Fe_3_O_4_ nanoparticles were precipitated in the surface and the internal of PGMA microspheres through the interfacial stripping precipitation method (see Figure 1b). Furthermore, a novel method in which carboxymethyl cellulose was bonded to the surface of magnetic microspheres by the EDC method was applied to prepare carboxyl magnetic microspheres (Figure 1c).

### 2.2. Characterization of the Magnetic Microspheres

The morphologies of microspheres were studied by SEM. It could be observed that the PGMA microspheres were mono-sized microspheres with very smooth surfaces, while the magnetic microspheres were relatively rough on the surface (Figure 2). The average diameter of the magnetic microspheres was 1.235 ± 0.017 μm according to the 100 microspheres selected from SEM images randomly, and the diameter of the Fe_3_O_4_ particles coated by sodium carboxymethyl cellulose on the microspheres was 30~50 nm. The size of these magnetic microspheres is relatively smaller compared with a similar polymerization method [19]. Generally, smaller diameter means larger specific surface area and greater adsorption capacity for the target.

The presence of the functional groups of PGMA and amino-PGMA microspheres was verified by FT-IR in Figure 3. PGMA microspheres were obtained by the polymerization of GMA monomer with DVB as cross-linker; therefore, epoxy groups and carbonyl group should distribute throughout the microspheres surface. The strong adsorption peak at 1727 cm^−1^ corresponds to C=O stretching vibration. Compared with Figure 3a, the characteristic bands at 848 cm^−1^, 908 cm^−1^, and 1250 cm^−1^ belong to the epoxy group which disappeared in Figure 3b, indicating that the epoxy groups transformed into an amino group after the amino modification.

Controlling the magnetite content of the microspheres is important for realizing the rapid response to external magnetic fields for efficient adsorption. TGA measurement showed that the main weight loss of all microspheres was in the range of 200~450 °C, indicating that the microspheres should have stable thermal performance in the adsorption condition (Figure 4A). By comparing the residual mass after full burning, it can be calculated that the magnetite content of amino magnetic spheres and carboxyl magnetic spheres is about 12%. The magnetic properties of the carboxyl magnetic microspheres were measured by vibrating sample magnetometry (VSM). The magnetization curve shows that the saturation magnetization of the microspheres reached 12.48 emu/g, while the residual magnetism and coercive force were almost zero (Figure 4B). It means the magnetic microspheres are of high enough magnetism such that they can be separated from a suspension quickly. Moreover, they are superparamagnetic, which could disperse in the solution uniformly when there is no magnetic field. Compared with previous reports, magnetic intensity lowering of sodium carboxymethyl cellulose coated magnetic decrease may be caused by the embedding of the coating.

The density of amino groups and carboxyl groups on corresponding microspheres measured by the titration method was 2.75 mmol/g and 1.32 mmol/g respectively. It could be inferred from the result that amino groups incompletely reacted with carboxymethyl cellulose, so both amino groups and carboxyl groups exist on the surface of carboxyl magnetic microspheres.

Investigation of the biological safety of magnetic microspheres is critical for biomolecular separation medium. Herein, human pulmonary epithelial cells were used to study the in vitro cytotoxicity of carboxyl magnetic microspheres solution and its supernatant measured using a water-soluble tetrazolium cell proliferation assay. It could be inferred from Figure 5 that the supernatant of suspension showed no cytotoxicity to the cell, and the carboxyl magnetic microspheres showed no cytotoxicity when their concentrations were lower than 10 μg/mg. Therefore, it could be said that the magnetic microspheres are potential targeting medicine carriers or cell separation carriers at low concentrations.

### 2.3. Binding Capability of Magnetic Microspheres

The adsorption capability of magnetic microspheres to proteins is mainly affected by the properties of the protein, the functional groups content on the surface of the microspheres, and the adsorption conditions. Generally, proteins are more likely to precipitate around their isoelectric point. It is comprehensible that pH value has a great effect on the adsorption of proteins by the microspheres. The maximum adsorption capacity of three proteins to amino magnetic microspheres is in close proximity to the isoelectric point (pI) in Figure 6a. Amino magnetic microspheres were positively charged while BSA was negatively charged at pH 5, which means BSA (pI 4.6) would be more easily absorbed on the magnetic surface of the microspheres due to its electrostatic interaction. The absorption mechanism of Hb (pI 7.0) was similar to BSA with the optimal amount of adsorption at pH 7. However, the adsorption for Cyt C (pI 10.65) was not very significant and the adsorption amount differences between each pH value were slight.

The carboxyl magnetic microsphere was negatively charged on the protein surface. Due to electrostatic adsorption, binding capacities of Hb to the microspheres were greater when the pH value was below 7. Since there were both carboxyl groups and amino groups existing on the carboxyl magnetic microsphere surface, the adsorption of carboxyl magnetic microspheres was more complicated than the adsorption of amino magnetic microspheres. Binding experimental results (Figure 6b) suggested the overall adsorption of BSA was not much with the maximum adsorption when the pH value was close to its isoelectric point; greater adsorption of Hb is observed in an acidic environment; maximum absorption of Cyt C is also near the isoelectric point. According to our experiment, the optimal adsorption conditions Hb, BSA, and Cyt C are pH 5, pH 7, and pH 9 for amino microspheres, pH 4, pH 3, and pH 11 for the carboxyl microspheres, respectively. The adsorption of Hb was also recorded in the concentration ranging from 0 to 10 mg/mL at 25 °C (Figure 7). With the increase of the initial concentration of Hb, the equilibrium adsorption amount generally increased and eventually became saturated.

The adsorption capacities of several magnetic microsphere adsorption materials were compared in Table 1. We can see that the adsorption capacity of the prepared microspheres is comparable to that of the magnetic microspheres prepared by the in situ method. Meanwhile, they have the advantages of simple preparation and low cost. The immunoaffinity magnetic microspheres are highly specific to the target protein, however, sacrificing part of the adsorption capacity. The magnetic microspheres prepared by the imprinting technology have ordered imprinting pores and excellent specificity for target protein, realizing an efficient adsorption rate, which is higher than immunomagnetic microspheres. In addition, the surface mesoporous structure and the enrichment effect of metal particles are also considered to be the factors that improve the ability to adsorb proteins.

The adsorption isotherm describes the distribution of the adsorbed molecules between the liquid phase and the solid phase when the adsorption process reaches equilibrium. To study the adsorption mechanism of amino magnetic microspheres and carboxyl magnetic microspheres to Hb, whose adsorption capacity is larger than that of other proteins, the isotherm data were analyzed based on the Langmuir and Freundlich models respectively. The expressions, adsorption constantsm and correlation coefficients of the Langmuir and Freundlich models at 25 °C were calculated and are presented in Table 2.

Where *c_e_* is the equilibrium concentration (mg/L), *q_e_* is the adsorption amount at equilibrium (mg/g), *K* is the Langmuir adsorption equilibrium constant, *q**_m_* is the Langmuir constant, which represents the saturated monolayer adsorption capacity (mg/g), *K_F_* is a Freundlich constant related to the adsorption capacity (mg/g), and *n* is a Freundlich adsorption equilibrium constant relevant to the adsorption intensity.

Comparing the adsorption constants and correlation coefficients (*R^2^*) of the Langmuir and Freundlich isotherms, it is suggested that the adsorption of the magnetic microspheres to Hb is in accordance with the Langmuir equation and the adsorption process is chemical monolayer adsorption. In Langmuir model, the maximum capacity q_mof the carboxyl microspheres in the Langmuir constant monolayer was 215.74 mg/g at 25 °C, indicating that the carboxyl magnetic microspheres have better adsorption capacity than the amino magnetic microspheres.

The kinetic curves (Figure 8) showed that the adsorption of the microspheres saturated quickly in 15 min. It is because electrostatic force between magnetic microspheres and proteins occurs mainly on the surface of the magnetic microspheres without the inward diffusion phenomenon. In the initial stage, there are a large number of active sites on the surface of the magnetic microspheres, and proteins were easily adsorbed. As the adsorption increases, the surface active sites of the magnetic microspheres decrease, and the adsorption becomes slower. Therefore, the microspheres would be more efficient for protein purification.

The adsorption data of the microspheres to Hb were respectively fitted to the pseudo first-order and pseudo-second-order kinetic models (Table 3). The correlation coefficients of the pseudo second-order kinetic model for amino magnetic microspheres and carboxyl magnetic microspheres were 0.9981 and 0.9911, and the maximum adsorption amounts obtained were 119.05 mg/g and 172.41 mg/g, respectively, which are consistent with the experimental results. It is suggested that the adsorption process is a pseudo second-order kinetic adsorption, which is consistent with the mentioned adsorption mechanism above.

Where $q_{e}$ and $q_{t}$ signify the amount adsorbed at equilibrium and at any time t, k is a Lagergren constant.

## 3. Materials and Methods

### 3.1. Materials

Glycidyl methacrylate (GMA) was obtained from TCI Company (Shanghai, China). Divinylbenzene was obtained from J&K chemical (Beijing, China). 1-Ethyl-3-(3-dimethyllaminopropyl) carbodiimide hydrochloride (EDC·HCl) and N-Hydroxysuccinimide (NHS) were purchased from Shanghai Medpep corporation (Shanghai, China). Bovine albumin (BSA), Hemoglobin from bovine blood (Hb), Cytochrome C (Cyt C) were obtained from Aladdin chemical corporation (Shanghai, China). Polyvinyl pyrrolidone (PVP K-30), 2,2′-azobis-(isobutyronitrile) (AIBN), ethylenediamine (EDA), anhydrous morpholine ethanesulfonic acid, and other chemicals were received from Beijing Chemical Factory (Beijing, China). Cell Counting Kit-8 (CCK-8) was received from Dojindo Laboratories, Kumamoto, Japan. Human pulmonary epithelial cells were purchased from InvivoGen (San Diego, CA, USA) and grown in Dulbecco’s modified Eagle’s medium (Sigma-Aldrich, St. Louis, MO, USA) containing 10% (*v/v*) fetal bovine serum, 50 units/mL penicillin, 50 mg/L streptomycin, 100 μg/mL normocin, and 10 μg/mL blasticidin. All chemicals were used without further treatment. Deionized water used in polymerization and characterization was distilled and purified by Aqua Pro (Chongqing, China).

### 3.2. Synthesis of PGMA Microspheres

PGMA microspheres are fabricated by dispersion polymerization method [24]. The polymerization was carried out under nitrogen in the three-necked flask equipped with a condenser. PVP K-30 (2.4 g) and GMA (8.0 g) dissolved in ethanol (67.0 g) was stirred at 300 rpm under nitrogen at room temperature for 15 min. After the initiator AIBN (0.16 g/5 g ethanol) was added, the polymerization was carried out at 70 °C for 2 h. Thereafter, Divinylbenzene (DVB, 0.24 g) was added into the flask smoothly, keeping the reaction going on for 5 h. Then the microspheres were centrifuged and washed with ethanol and water several times and dried under vacuum.

### 3.3. Synthesis of Amino Magnetic Microspheres

The amino magnetic microspheres were synthesized according to the reported method [15]. The dry PGMA microspheres (2.0 g) were added into a mixture of ethylene diamine (EDA, 50 mL) and water (50 mL) while stirring at 80 °C for 6 h. The microspheres were centrifuged and washed with water, and then dried under vacuum. The EDA functionalized microspheres (1.0 g) were added into water (100 mL), which was cooled to 0 °C under nitrogen for 30 min. Afterwards, FeCl_3_·6H_2_O (0.41 g) and FeSO_4_·7H_2_O (0.24 g) dissolved in water (10 mL) were added to the mixture respectively, and stirred for 3 h below 5 °C. After adding the ammonia solution (10 mL) smoothly, the ice bath was removed and the temperature was raised to 80 °C for 1.5 h. The resulting microspheres were centrifuged and washed with 0.5 M HCl three times and followed by pure water. The magnetic microspheres were dried by lyophilization and reserved.

### 3.4. Synthesis of Carboxyl Magnetic Microspheres

The amino magnetic microspheres were modified with sodium carboxymethyl cellulose to the synthesis of the amino-microspheres. EDC (1.94 g) and NHS (0.58 g) were dissolved in MES solution (100 mL, 0.1 M) together with sodium carboxymethyl cellulose solution (100 mL, 2.5 g/L). Then the dry amino magnetic microspheres (1.0 g) were added and stirred at room temperature for 2 h. Finally, the microspheres were washed with pure water and dried by lyophilization.

### 3.5. Characterizations of the Magnetic Microspheres

The morphology of magnetic microspheres was observed by scanning electron microscopy (SEM, S-4800, HITACHI, Tokyo, Japan). The sample powders were sputter-coated with gold before examination. The magnetic properties of magnetic microspheres were measured by a vibrating sample magnetometer (VSM, 9600-1, LDJ Electronics, Troy, MI, USA) at room temperature. TGA was performed with a thermal gravimetric analyzer (DTG-60H, Shimadzu, Kyoto, Japan) in the temperature range from room temperature to 800 °C with a scanning rate of 10 °C/min under nitrogen stream. The presence of certain functional groups was detected by Fourier Transform infrared spectrometer (FT-IR, ALPHA, Bruker, Billerica, MD, USA). The densities of amino groups and carboxyl groups on the microspheres were measured by a titration method.

### 3.6. Cytotoxicity Test of the Carboxyl Magnetic Microspheres

The cytotoxicity of the carboxyl magnetic microspheres was investigated using a CCK-8 method in vitro. The 96-well plates were seeded with a suspension of 5000 human pulmonary epithelial cells for 24 h to allow the cells to adhere. Then serial dilutions of carboxyl magnetic microspheres solution, the supernatant and medium alone (control) were added into the wells. After incubation at 37 °C for 24 h in an atmosphere of 5% CO_2_, 10 μL CCK-8 solution was added to each well and the cells were incubated for another three hours. Absorbance at 450 nm was determined using a microplate reader using a microplate reader (MTP-880 Lab, Corona Electric, Ibaraki, Japan). Cytotoxicity was expressed as a percentage of viable cells compared with untreated control ones.

### 3.7. Binding Experiment

The binding properties of magnetic anion-exchange microspheres to the proteins were studied by HPLC with a diode array detector and the C8 column at 40 °C. The standard curves and adsorption capability for the proteins were measured with the corresponding conditions. For BSA, the mobile phase was acetonitrile/water (2/8–8/2, *v/v*), using a linear gradient elution at the wavelength of 280 nm, and the injection volume was 10 μL. For Hb, the mobile phase was acetonitrile/water (5/5, *v/v*), using isocratic elution at the wavelength of 400 nm, and the injection volume was 20 μL. For Cyt C, the mobile phase was acetonitrile/water (3/7–5/5, *v/v*), using linear gradient elution at the wavelength of 400 nm, and the injection volume was 20 μL.

The adsorption of proteins by magnetic microspheres was carried out in phosphate buffers (100 mM) of different pH values ranging from 3–11, adjusted with phosphoric acid solution or sodium hydroxide solution. The following experiment was performed in triplicate. The dry magnetic polymer microspheres (5 mg) were dispersed in 1 mL buffer solution followed by the adsorption experiment while the initial concentration of protein was determined from 0 to 10 mg/mL and the equilibrium time was 60 min. In the adsorption kinetics experiment, protein solution with initial concentration (BSA 5 mg/mL, Hb 5 mg/mL, Cyt C 1 mg/mL) was added, and the mixture was incubated at 25 °C for different times (0~60 min) respectively. Then the tubes were placed in the magnetic separation rack for 2 min, and the supernatant was extracted carefully for HPLC detection.

The protein binding quantity *q* (mg/g) of magnetic microspheres could be calculated from Formula (1).
(1)$$q=\frac{\left(C_{0}-C\right)\times V}{W}$$
where *C*_0_ and *C* are the protein concentrations (mg/mL) before and after adsorption; *V* is the volume of protein solution (mL); *W* is the weight of magnetic microspheres (g).

## 4. Conclusions

A new process to obtain the carboxymethyl cellulose surface-coated magnetic polymer microspheres by EDC method was performed in this study. The superparamagnetism and no significant cytotoxicity of the magnetic microspheres attribute to their potential application in vivo. The adsorption capacity of three proteins (BSA, Hb, and Cyt C) on amino magnetic microspheres and carboxyl magnetic microspheres was evaluated, wherein maximum adsorption capacity of Hb on carboxyl magnetic microspheres reached 215.74 mg/g within sufficient binding time at appropriate pH value. However, further studies based on the increase of the stability of magnetic microspheres, specific adsorption of a certain protein, and desorption of protein are required. This paper provides an idea for the preparation of magnetic microspheres for protein separation, which is expected to be a fast and efficient new way of protein separation in the future.

## Figures and Tables

**Figure 1 ijms-23-04963-f001:**
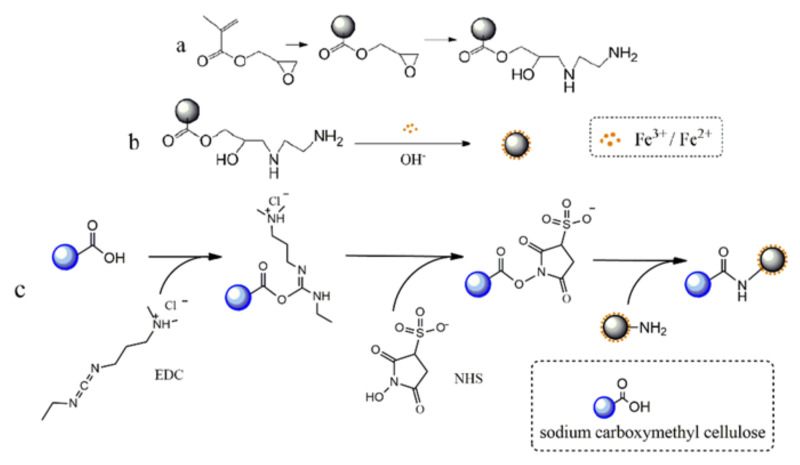
Schematic diagram of the preparation of magnetic anion-exchanged microspheres. (**a**) Preparation and amino modification of PGMA Polymer Microspheres. (**b**) Synthesis of amino magnetic microspheres. (**c**) Carboxyl coating on the surface of amino magnetic microspheres.

**Figure 2 ijms-23-04963-f002:**
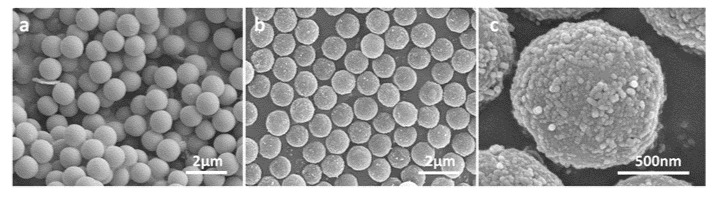
SEM images of microspheres. (**a**) PGMA microspheres. (**b**) Carboxyl magnetic microspheres. (**c**) The close-up of carboxyl magnetic microspheres.

**Figure 3 ijms-23-04963-f003:**
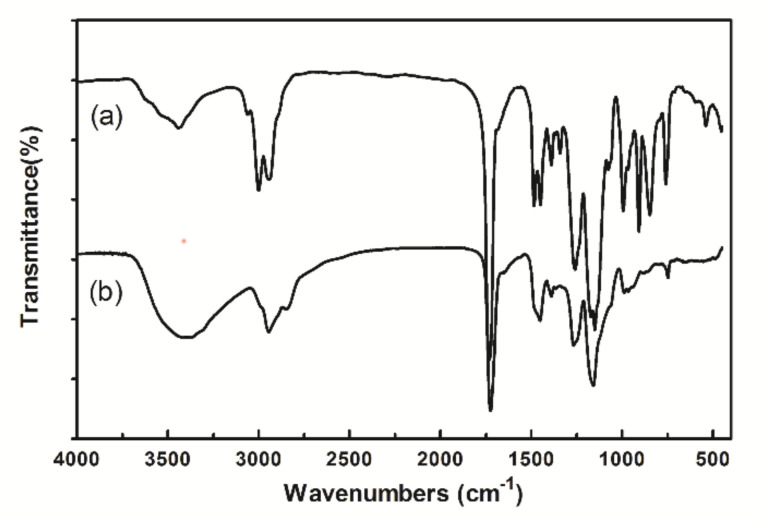
IR spectrum of microspheres. (**a**) PGMA microspheres. (**b**) Amino microspheres.

**Figure 4 ijms-23-04963-f004:**
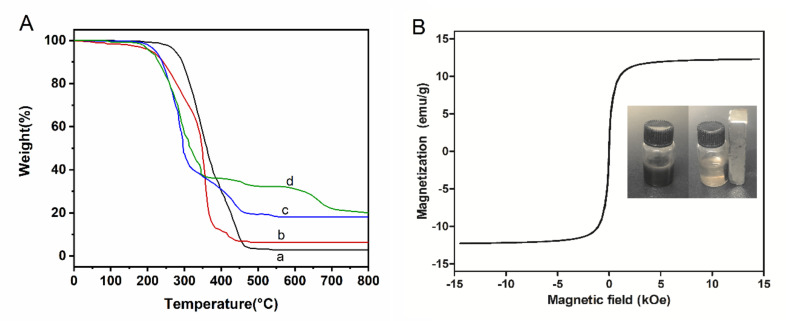
(**A**) TGA of (a) PGMA microspheres, (b) Amino microspheres, (c) Amino magnetic microspheres, and (d) Carboxyl magnetic microspheres. (**B**) Hysteresis curve of carboxyl magnetic microspheres. The inset shows photos of the carboxyl magnetic microsphere in aqueous solution without (left) and with (right) a magnet.

**Figure 5 ijms-23-04963-f005:**
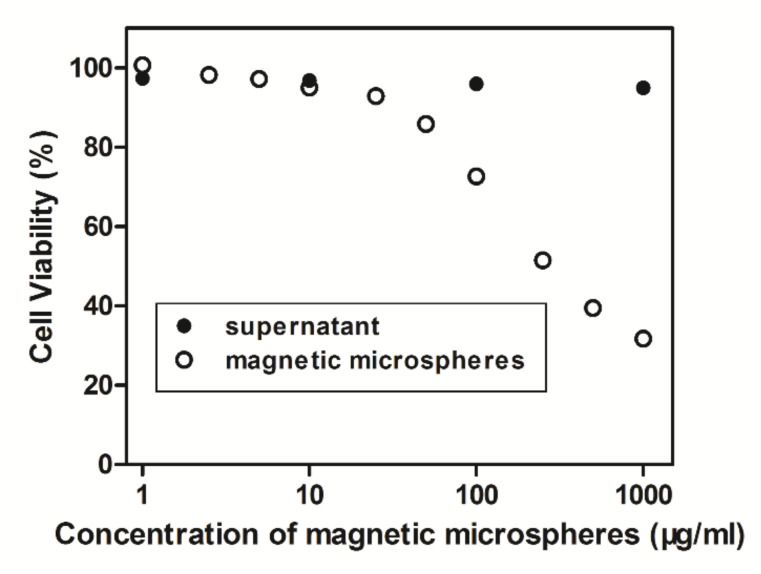
Relation curve of cell viability with the concentration of magnetic microspheres.

**Figure 6 ijms-23-04963-f006:**
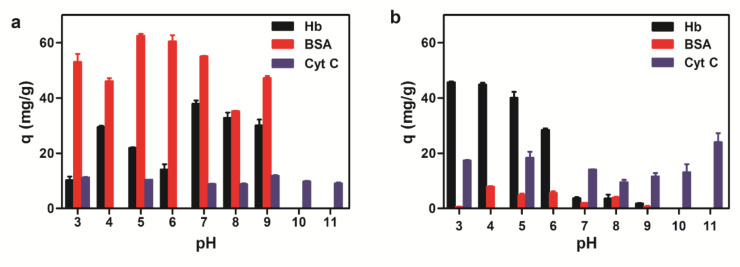
Adsorption capacity of three proteins to magnetic microspheres with different pH values. (**a**) Amino magnetic microspheres. (**b**) Carboxyl magnetic microspheres.

**Figure 7 ijms-23-04963-f007:**
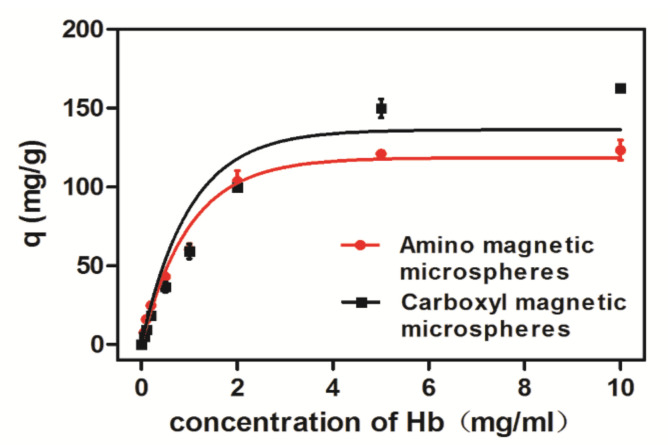
Adsorption isotherm of magnetic microspheres on Hb at 25 °C.

**Figure 8 ijms-23-04963-f008:**
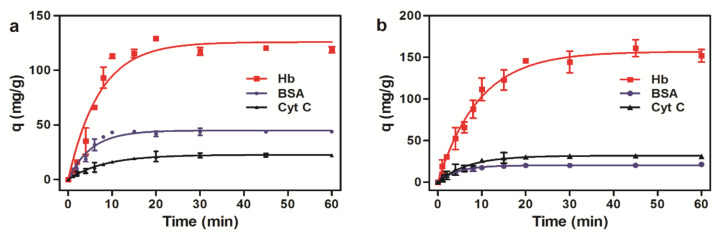
Adsorption kinetics of three proteins to magnetic microspheres. (**a**) Amino magnetic microspheres. (**b**) Carboxyl magnetic microspheres.

**Table 1 ijms-23-04963-t001:** Comparison of the adsorption capacity of several magnetic microspheres for proteins.

Magnetic Microspheres	Protein	Maximum Adsorption Capacity (mg/g)
Thepreparedmagnetmicrospheres	Hb	217
Chitosan-based magnetic beads byin situmethod [20]	BSA	240.5
Cu^2+^-cooperated magnetic imprinted nanomaterial [21]	Hb	116.3
surface-imprinted polyvinyl alcohol microspheres [22]	papain	44
magnetic immunoaffinity beads by dispersion polymerization [23]	Anti-Tf	2.0
Fe_3_O_4_@PMAA@Ni microspheres with flower-like Ni nanofoams [7]	Hb	2660

**Table 2 ijms-23-04963-t002:** The adsorption isotherm parameters of Hb by amino magnetic microspheres and carboxyl magnetic microspheres.

Adsorbents	Langmuir Adsorption Isotherm $\frac{c_{e}}{q_{e}}=\frac{c_{e}}{q_{m}}+\frac{1}{Kq_{m}}$			Freundlich Adsorption Isotherm $\ln q_{e}=\ln K_{F}+\frac{lnc_{e}}{n}$		
	$q_{m}\;(\mathrm{mg}/g)$	*K*	*R* ^2^	$K_{F}\;(\mathrm{mg}/g)$	1/*n*	*R* ^2^
amino magnetic microspheres	131.27	1.4837	0.9966	52.55	0.5342	0.9427
carboxyl magnetic microspheres	215.74	0.4282	0.9997	48.41	0.6603	0.9750

**Table 3 ijms-23-04963-t003:** Adsorption kinetics parameters of HB by amino magnetic microspheres and carboxyl magnetic microspheres.

	Amino Magnetic Microspheres			Carboxyl Magnetic Microspheres		
	k	$q_{e}/(\mathrm{mg}/g)$	*R^2^*	k	$q_{e}/(\mathrm{mg}/g)$	*R* ^2^
Lagergren first-order rate kinetics $\ln\left(q_{e}-q_{t}\right)=\ln q_{e}-kt$	0.2660	166.05	0.9209	0.1143	153.33	0.9767
Lagergren second-order rate kinetics $\frac{t}{q_{t}}=\frac{1}{kq_{e}^{2}}+\frac{t}{q_{e}}$	2.352	119.05	0.9981	0.0010	172.41	0.9911

## Data Availability

Not applicable.

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
