# Peer review of "Mono-Sized Anion-Exchange Magnetic Microspheres for Protein Adsorption"

_ijms, 2022, doi:10.3390/ijms23094963_

Round 1

Reviewer 1 Report

The paper dealing with new way of preparation magnetic spheres that can be further functionalized and used for selection of various proteins with its scope fall within the scope of the journal and can be potentially considered for its publication. The paper is well written and easy to read, however, in my opinion it can be further improve.

1) Is it somehow possible to estimate the amount of magnetic particles on the surface of the PGMA microspheres (TGA technique). Is it possible to further control the amount of magnetic particles or the amount of precursors is calculated to fully occupy the surface of the PGMA particles?

2) The prepared magnetic microspheres appear no cytotoxicity and concentration lower than 10 mg/ml. Could you please specify the concentration of magnetic polymer microspheres during binding experiment.

3) In my opinion the paper strongly misses comparison with conventional systems within the discussion of results. Whether the prepared systems exceeds the state-of-the-art systems.

Author Response

Response to Reviewer 1 Comments

Comments and Suggestions for Authors

The paper dealing with new way of preparation magnetic spheres that can be further functionalized and used for selection of various proteins with its scope fall within the scope of the journal and can be potentially considered for its publication. The paper is well written and easy to read, however, in my opinion it can be further improve.

Response: Thank you very much for reviewing our manuscript “Mono-sized anion-exchange magnetic microspheres for protein adsorption” (Manuscript ID: ijms-1657546). Your valuable comments and suggestions greatly helped us to improve our manuscript. We have carefully revised our manuscript according to your suggestions, and point-by-point answered the questions as following.

Point 1: Is it somehow possible to estimate the amount of magnetic particles on the surface of the PGMA microspheres (TGA technique). Is it possible to further control the amount of magnetic particles or the amount of precursors is calculated to fully occupy the surface of the PGMA particles?

Response 1:

According to your suggestion, TGA test of the products during preparation was carried out and corresponding changes in the manuscript are as follows.

The following sentences were added in the Abstract “The obtained magnetic microspheres showed a uniform particle diameter of 1.235 μm in average and a good spherical shape with a saturation magnetic intensity of 12.48 emu/g by VSM and 12% magnetite content by TGA.” Results and discussion Section (2.2. Characterization of the magnetic microspheres) “Controlling the magnetite content of the microspheres is important for realize the rapid response to external magnetic fields for efficient adsorption. TGA measurement showed that the main weight loss of all microspheres was in the range of 200 ~ 450°C, indicating that the microspheres should have stable thermal performance in the ad-sorption condition (Figure 4A). By comparing the residual mass after full burning, it can be calculated that the magnetite content of amino magnetic spheres and carboxyl magnetic spheres is about 12%.”and Figure 4. Materials and Methods section (3.5. Characterizations of the magnetic microspheres) “TGA was performed with thermal gravimetric analyzer (DTG-60H, Shimadzu) in the temperature range from room temperature to 800 °C with a scanning rate of 10 °C /min under nitrogen stream.”

Theoretically, it is possible to control the number and size of magnetic particles on the amino microspheres by adjusting the formulation of the pre-polymerization solution and the reaction conditions by in situ method. ( ACS Omega. 2020, 5, 15, 8839–8846. J. Mater. Chem., 2009, 19, 3538–3545. Macromol. Res.2012. 20, 1211–1218.) So far as I know, the maximum magnetite content of magnetic particle-polymer composites is 86%, according to the work of Gu (J. Polym. Sci., Part A: Polym. Chem.  2007, 45, (22), 5285-5295). For the prepared microspheres, we tested the magnetic properties of the carboxyl magnetic microsphere samples by VSM, and the saturation magnetization was 12.48 emu/g. It is enough for the magnetic microspheres to be separated quickly and efficiently from the mixed liquid with a magnet. Conversely, too many magnetic particles may lead to agglomeration or shedding from our microspheres.

Point 2: The prepared magnetic microspheres appear no cytotoxicity and concentration lower than 10 mg/ml. Could you please specify the concentration of magnetic polymer microspheres during binding experiment.

Response 2:

We are grateful to you for pointing out this problem. Dried magnetic microspheres were used as adsorbents in the binding experiments, and 5 mg of magnetic microspheres were added to 1 mL of protein solution. The concentration of microspheres in the solution is 5 mg/mL. According to our experiment, the prepared magnetic microspheres appeared no cytotoxicity when its concentration was lower than 10 mg/mL.

The following sentences were modified or added in the Materials and Methods section section. (3.7. Binding experiment) “The dry magnetic polymer microspheres (5 mg) were dispersed in 1 mL buffer solution followed by the adsorption experiment while the initial concentration of protein was determined from 0 to 10 mg/mL and the equilibrium time was 60 min.”

Point 3: In my opinion the paper strongly misses comparison with conventional systems within the discussion of results. Whether the prepared systems exceeds the state-of-the-art systems.

Response 3:

Considering the reviewer’s suggestion, the comparison of the adsorption capacity of several magnetic microspheres for proteins was added in manuscript.

We have added the following sentences and table in the Results and discussion.( 2.3. Binding capability of magnetic microspheres) “The adsorption capacity of several magnetic microsphere adsorption materials was compared in Table 1. We can see that the adsorption capacity of the prepared microspheres is comparable to that of the magnetic microspheres prepared by the in-situ method. Meanwhile, they have the advantages of simple preparation and low cost. The immunoaffinity magnetic microspheres are highly specific to the target protein, however, sacrificing part of the adsorption capacity. The magnetic microspheres prepared by the imprinting technology have ordered imprinting pores and excellent specificity for target protein, realizing an efficient adsorption rate, which is higher than immunomagnetic microspheres. In addition, the surface mesoporous structure and the enrichment effect of metal particles is also considered to be the factors that improve the ability to adsorb proteins.”

Table 1. Comparison of the adsorption capacity of several magnetic microspheres for proteins

Magnetic microspheres

Protein

Maximum adsorption capacity (mg/g)

The prepared magnet microspheres

Hb

217

Chitosan-based magnetic beads by in situ method[20]

BSA

240.5

Cu2+-cooperated magnetic imprinted nanomaterial[21]

Hb

116.3

surface-imprinted polyvinyl alcohol microspheres[22]

papain

44 

magnetic immunoaffinity beads by dispersion polymerization[23]

Anti-Tf

2.0

Fe3O4@PMAA@Ni microspheres with flower-like Ni nanofoams[7]

Hb

2660

Besides, some content of the manuscript has been modified according to reviewers' comments. The expression and grammar have been reviewed and revised carefully in the manuscript

We hope all the above answers will satisfy you. If you still have any suggestions, please feel free to tell us. We will carefully revise our manuscript again. Thank you sincerely again for your time and suggestions.

Reviewer 2 Report

The article submitted to IJMS concerns the current subject of the synthesis and application of magnetic nanospheres as a support capable of binding proteins. The topic is quite popular, and for the work to be attractive for the reader of a journal such as IJMS, it should contain essential news elements which is missing in the submitted article.

The method of obtaining microspheres is known, and the polymer used for synthesis has been well researched and described. In addition, the obtained binding results are predictable because the process was carried out at different pH values, which caused changes in charges both on the surface of the microspheres and protein structures.

The characteristics of the materials obtained are also relatively poor. No thermal characteristics of the material, no precise measurement of the size of the microspheres and no TEM images that would allow for a better "view" of the material. It is also required to perform a nitrogen adsorption-desorption isotherm with porous materials for the determination.

The work is poorly written, the sentences are repeated, and the drawings showing the results of the analyzes (Fig. 3,4 ...) are of poor quality, making them questionable. I believe that the submitted work cannot be published in IJMS.

Author Response

Response to Reviewer 2 Comments

 The article submitted to IJMS concerns the current subject of the synthesis and application of magnetic nanospheres as a support capable of binding proteins. The topic is quite popular, and for the work to be attractive for the reader of a journal such as IJMS, it should contain essential news elements which is missing in the submitted article.

Thank you very much for reviewing our manuscript “Mono-sized anion-exchange magnetic microspheres for protein adsorption” (Manuscript ID: ijms-1657546). Your valuable comments and suggestions greatly helped us to improve our manuscript. We have carefully revised our manuscript according to your suggestions, and point-by-point answered the questions as following.

 Point 1: The method of obtaining microspheres is known, and the polymer used for synthesis has been well researched and described. In addition, the obtained binding results are predictable because the process was carried out at different pH values, which caused changes in charges both on the surface of the microspheres and protein structures.

Response 1: The reviewer has commented that the method synthesis and the polymer materials have been well researched, and the adsorption results are predictable. In this manuscript, PGMA-based magnetic microspheres were prepared and their separation capability was studied. Herein, the innovation was that The EDC method was used to coat carboxymethyl cellulose on the surface of magnetic microspheres to obtain magnetically stable carboxyl magnetic polymer microspheres. Protein separation is mainly based on five principles: molecular size, solubility, charge, adsorption properties, biological affinity for ligand molecules, etc. Among them, the effect of charge changes affected by pH is a relatively simple method to achieve protein adsorption. However, further study is needed on increasing functional groups on surface, enhancing the stability of magnetic microspheres and specific protein adsorption and separation.

 Point 2: The characteristics of the materials obtained are also relatively poor. No thermal characteristics of the material, no precise measurement of the size of the microspheres and no TEM images that would allow for a better "view" of the material. It is also required to perform a nitrogen adsorption-desorption isotherm with porous materials for the determination.

Response 2:

Thank the reviewers for these precious comments and suggestions. To the best of our knowledge, TGA is extremely good method to measure the thermal stability and the weight contribution of magnet nanoparticles in polymer microspheres. So TGA test of the products during preparation was carried out and corresponding changes in the manuscript are as follows.

The following sentences were added in the Abstract “The obtained magnetic microspheres showed a uniform particle diameter of 1.235 μm in average and a good spherical shape with a saturation magnetic intensity of 12.48 emu/g by VSM and 12% magnetite content by TGA.” Results and discussion Section (2.2. Characterization of the magnetic microspheres) “Controlling the magnetite content of the microspheres is important for realize the rapid response to external magnetic fields for efficient adsorption. TGA measurement showed that the main weight loss of all microspheres was in the range of 200 ~ 450°C, indicating that the microspheres should have stable thermal performance in the ad-sorption condition (Figure 4A). By comparing the residual mass after full burning, it can be calculated that the magnetite content of amino magnetic spheres and carboxyl magnetic spheres is about 12%.”and Figure 4. Materials and Methods section (3.5. Characterizations of the magnetic microspheres) “TGA was performed with thermal gravimetric analyzer (DTG-60H, Shimadzu) in the temperature range from room temperature to 800 °C with a scanning rate of 10 °C /min under nitrogen stream.”

We used SEM instead of TEM to characterize the microsphere morphology because the particle size of the prepared microspheres is large enough to obtain accurate and clear topographic pictures with the resolution of SEM. As mentioned in the manuscript, “The average diameter of the magnetic microspheres is 1.235±0.017 μm according to the 100 microspheres selected from SEM images randomly, and the diameter of the Fe3O4 particles coated by sodium carboxymethyl cellulose on the microspheres is 30~50 nm.” We also can get distinguishable pictures from Figure 1.

After coating with sodium carboxymethyl cellulose, the pores on the surface of the magnet PGMA microspheres formed by the Pore forming agent are covered. In the protein adsorption application, the carboxyl and amino groups on the surface of the magnet microspheres play a major role, so we do not think it is necessary to get the data of specific surface area and porosity of the microspheres from BET test.

Point 3: The work is poorly written, the sentences are repeated, and the drawings showing the results of the analyzes (Fig. 3,4 ...) are of poor quality, making them questionable.

Response 3: The expression and grammar have been reviewed and revised carefully in the manuscript, several repeated sentences were deleted to make the manuscript condense and easily understandable.

We hope all the above answers will be helpful for your re-evaluation. If you still have any suggestions, please feel free to tell us. We will carefully revise our manuscript again. Thank you sincerely again for your time and suggestions.
